# A loss framework for calibrated anomaly detection

**Aditya Krishna Menon**
Australian National University*
aditya.menon@anu.edu.au

**Robert C. Williamson**
Australian National University
bob.williamson@anu.edu.au

## Abstract

Given samples from a distribution, anomaly detection is the problem of determining if a given point lies in a low-density region. This paper concerns *calibrated* anomaly detection, which is the practically relevant extension where we additionally wish to produce a *confidence* score for a point being anomalous. Building on a classification framework for standard anomaly detection, we show how minimisation of a suitable *proper loss* produces density estimates only for anomalous instances. These are shown to naturally relate to the pinball loss, which provides implicit quantile control. Finally, leveraging a result from point processes, we show how to efficiently optimise a special case of the objective with kernelised scores. Our framework is shown to incorporate a close relative of the one-class SVM as a special case.

## 1 Calibrated anomaly detection

Given a set of instances with some systematic pattern (e.g., images of household objects), anomaly detection is informally understood as the problem of identifying if a particular instance deviates from this pattern (e.g., an image of a tree). There are several formalisations of this notion, each of which have led to many different algorithms (see, e.g., Chandola et al. [2009], Pimentel et al. [2014] for some recent surveys). For example, in the *density sublevel set* formulation of anomaly detection, given samples from a probability distribution, we wish to determine if an instance belongs to the induced low-density region of the input space [Ripley, 1996, pg. 24–25], [Steinwart et al., 2005], [Chandola et al., 2009, Section 5.2].

A practically relevant extension of this problem, which we term *calibrated* anomaly detection, is where we additionally wish to produce a *confidence* score for an instance being anomalous [Gao and Tan, 2006, Sotiris et al., 2010, Kriegel et al., 2011]. Intuitively, this confidence reflects the instance's level of abnormality, i.e., the *relative* value of its density compared to the other instances. Such information is valuable for subsequent decision making using these predictions [Gao and Tan, 2006].

In this paper, we present a loss function framework for calibrated anomaly detection. This builds upon an extant classification framework for density sublevel set problem [Steinwart et al., 2005], which related the latter to binary classification of samples from the observed distribution against a known "background" distribution. Our contributions are to extend this framework in three ways:

(**C1**) we show that minimising a class of *proper* losses asymptotically estimates the density for anomalous instances, while ignoring the density for non-anomalous instances

(**C2**) we show how to obtain (implicit) *quantile control* by connecting these proper losses to a generalised pinball loss for quantile elicitation [Ehm et al., 2016, Equation 5]

(**C3**) for a specific family of losses, we show how efficient optimisation can be performed with *kernelised* scores using an idea from the field of point processes [Flaxman et al., 2017].

**C2** & **C3** are also applicable to standard anomaly detection, and thus may be of independent interest.

| Method | Bayes-optimal? | Calibration? | Quantile control? | Tractable optimisation? |
|---|---|---|---|---|
| Heuristic sampling [Fan et al., 2001] | × | × | × | ✓ |
| Cost-sensitive loss [Steinwart et al., 2005] | ✓ | × | × | ∼ |
| Minimum volume set [Scott and Nowak, 2006] | ✓ | ✓ | ✓ | × |
| One-class SVM [Schölkopf et al., 2001] | ∼ | ∼ | ✓ | ✓ |
| Ours | ✓ | ✓ | ✓* | ✓ |
| | §4.2 | §4.2 | §5 | §6 |

Table 1: Comparison of various loss-based approaches to anomaly detection. For each method, we indicate whether they asymptotically produce a Bayes-optimal solution; can produce calibrated anomaly scores; offer quantile control; and afford tractable optimisation. The approach of Steinwart et al. [2005] is tractable only when sampling to approximate a high-dimensional integral. For the one-class SVM, Bayes-optimality and calibration are only achieved with certain approximations; see §6.2. For our method, we use a qualified ✓* since our quantile control is implicit; see §5.

Broadly, our aim in this paper is not to propose a new "best" method for (calibrated) anomaly detection. Rather, akin to Steinwart et al. [2005], we wish to explicate what precisely the target object for the problem is, and characterise the family of losses which yield this object upon risk minimisation. Optimisation-specific considerations are then built upon this foundation, allowing for a separation of statistical and computational concerns.

As an illustration, a special case of our framework closely resembles the one-class SVM (OC-SVM) [Schölkopf et al., 2001]. This algorithm has previously been noted to produce estimates of the tail density in a certain limiting regime, owing to the regulariser acting as both a model penalty, and a risk approximator [Vert and Vert, 2006]. Our framework allows one to separate these statistical and computational roles, and ascribe their influence to individual terms in the OC-SVM objective. Table 1 compares our framework to this, and various other loss-based approaches to anomaly detection.

## 2 Background on anomaly detection

Fix a probability distribution $P$ over a measurable space $\mathcal{X}$ (typically $\mathbb{R}^n$ with the Borel sigma-algebra), and a reference measure $\mu$ (typically Lebesgue) for which the density $p \doteq \frac{\mathrm{d}P}{\mathrm{d}\mu}$ exists.

**Standard anomaly detection**. Given samples from $P$, there are two ways we can pose low-density anomaly detection. The first is the *density sublevel set* formulation [Ripley, 1996, pg. 24–25], [Steinwart et al., 2005]. Here, we are given $\alpha > 0$, and wish to produce $c \colon \mathcal{X} \to \{\pm 1\}$ satisfying

$$c(x) = 2 \cdot [\![ p(x) > \alpha ]\!] - 1. \qquad (1)$$

That is, instances with low density are deemed to be anomalous. For simplicity, we assume $P(\{x \in \mathcal{X} \colon p(x) = \alpha\}) = 0$, and thus disregard ties. The second is the *minimum volume set* or *p-value* formulation [Scott and Nowak, 2006, Zhao and Saligrama, 2009, Chen et al., 2013]. Here, we are given $q \in (0, 1)$, and wish to produce $c \colon \mathcal{X} \to \{\pm 1\}$ satisfying

$$c(x) = 2 \cdot [\![ p(x) > \alpha_q ]\!] - 1,$$

where $\alpha_q$ is the $q$th quantile of the random variable of density scores $\mathsf{P} \doteq p(\mathsf{X})$, where $\mathsf{X} \sim P$. That is, instances with low density *relative to other instances* are deemed to be anomalous. The appeal of this is one does not need to *a priori* know the range of density values to pick a suitable threshold $\alpha$.

Both formulations of anomaly detection have been the subject of considerable study, and algorithmic development [Chandola et al., 2009]. These include the one-class SVM [Schölkopf et al., 2001, Tax and Duin, 2004], neighbourhood-based approaches [Breunig et al., 2000, Zhao and Saligrama, 2009], and classification-based approaches [Fan et al., 2001, Steinwart et al., 2005, Cheema et al., 2016].

**Calibrated anomaly detection**. In *calibrated anomaly detection*, we wish to output a confidence in an instance being anomalous, based on the density quantiles: we are given $q \in (0, 1)$, and wish to produce $\eta \colon \mathcal{X} \to [0, 1]$ satisfying

$$\eta(x) \in \begin{cases} \{F_{\mathsf{P}}(p(x))\} & \text{if } p(x) < \alpha_q \\ [q, 1] & \text{if } p(x) > \alpha_q, \end{cases} \qquad (2)$$

where $F_{\mathsf{P}}$ denotes the cumulative distribution function of $\mathsf{P} = p(\mathsf{X})$. Intuitively, $F_{\mathsf{P}}(p(x))$ is the density quantile corresponding to $p(x)$, so that $p(x) > \alpha_q \iff F_{\mathsf{P}}(p(x)) > q$. The goal is to model this quantile only for the anomalous instances, i.e., those with $p(x) < \alpha_q$; for non-anomalous instances, we need only stipulate that the outputs are larger than all anomalous ones.

Calibrated anomaly detection may be contrasted to the problem of density estimation [Wasserman, 2006, Chapter 6], which seeks to accurately model the density $p$, rather than $F_{\mathsf{P}} \circ p$; since the range of the density will be unknown *a priori*, the raw values will not be obviously interpretable as a measure of confidence. It may also be contrasted to the problem of anomaly ranking [Clémençon and Jakubowicz, 2013, Goix et al., 2015], which seeks to produce *any* strictly increasing transformation of $p$; the resulting scores thus may not be meaningful probabilities. Nonetheless, given a scorer $f \colon \mathcal{X} \to \mathbb{R}$ which preserves the order of the densities, and instances $\{x_n\}_{n=1}^N \sim P^N$, one may compute the empirical $p$-value [Zhao and Saligrama, 2009, Chen et al., 2013]

$$\hat{\eta}(x; \{x_n\}) \doteq \frac{1}{N} \sum_{n=1}^N [\![ f(x_n) < f(x) ]\!]. \tag{3}$$

## 3    Losses for density (sublevel-set) estimation

We review a result of Steinwart et al. [2005], which showed how to cast density sublevel set estimation as loss minimisation. We then show how by tweaking the loss, we can cast full density estimation in the same framework. This will motivate tuning the loss to interpolate between these problems.

### 3.1    A loss framework for density (sublevel set) estimation

Let $\ell \colon \{\pm 1\} \times \mathbb{R} \to \mathbb{R}$ be a binary label loss, such as the logistic loss $\ell(y, v) = \log(1 + e^{-yv})$. Let $f \colon \mathcal{X} \to \mathbb{R}$ be a measurable, real-valued scorer. For $P, \mu$ as in §2, suppose $\ell(+1, f(\cdot))$ is $P$-integrable, and $\ell(-1, f(\cdot))$ is $\mu$-integrable. Define the *risk* of $f$ as

$$R(f; P, \mu, \ell) \doteq \mathop{\mathbb{E}}_{\mathsf{X} \sim P} [\ell(+1, f(\mathsf{X}))] + \int_{\mathcal{X}} \ell(-1, f(x)) \, \mathrm{d}\mu(x). \tag{4}$$

Where clear from context, we drop the dependence of the risk on $(P, \mu, \ell)$. When $\mu(\mathcal{X}) < +\infty$, $\mu$ is a scaled probability measure, and $R(f)$ is the risk for a binary classification problem of distinguishing "positive" observations (drawn from $P$) from a "negative" background (drawn from $\mu$). We now see that minimising $R(f)$ can be used for density (sublevel) set estimation, by appropriately choosing $\ell$.

**Density sublevel set estimation as classification**. Steinwart et al. [2005] observed that for suitable *cost-weighted* losses, one may use (4) to solve density sublevel set estimation. Specifically, let $\ell \colon \{\pm 1\} \times \mathbb{R} \to \mathbb{R}$ be any classification-calibrated loss (e.g., hinge or logistic) in the sense of [Zhang, 2004, Bartlett et al., 2006]. Given a cost parameter $c \in (0, 1)$, define the cost-weighted loss

$$(\forall v \in \mathbb{R}) \, \ell^{(c)}(+1, v) \doteq (1 - c) \cdot \ell(+1, v) \qquad \ell^{(c)}(-1, v) \doteq c \cdot \ell(-1, v). \tag{5}$$

As an example, for the 0-1 loss $\ell_{01}$, we obtain the *cost-sensitive loss* $\ell_{01}^{(c)}$.

It turns out that minimising (4) with $\ell^{(c)}$ yields density sublevel set estimates. One can see this by studying the *Bayes-optimal scorers* for the risk, i.e., the theoretical minimisers of the risk over the set $\mathcal{M}(\mathcal{X}, \mathbb{R})$ of all measurable scorers. The following is a trivial generalisation of [Steinwart et al., 2005, Proposition 5], which was for $\ell$ being the 0-1 loss $\ell_{01}$.

**Proposition 1:** *Pick any $\alpha > 0$, and classification-calibrated $\ell$. For $c_\alpha \doteq \frac{\alpha}{1+\alpha}$ and $\ell^{(c_\alpha)}$ per (5), let*

$$f^* \in \mathop{\mathrm{Argmin}}_{f \in \mathcal{M}(\mathcal{X}, \mathbb{R})} R(f; P, \mu, \ell^{(c_\alpha)})$$

$$= \mathop{\mathrm{Argmin}}_{f \in \mathcal{M}(\mathcal{X}, \mathbb{R})} \mathop{\mathbb{E}}_{\mathsf{X} \sim P} [\ell(+1, f(\mathsf{X}))] + \alpha \cdot \int_{\mathcal{X}} \ell(-1, f(x)) \, \mathrm{d}\mu(x).$$

*Then, for $\mu$-almost every $x \in \mathcal{X}$, $f^*(x) > 0 \iff p(x) > \alpha$.*

Proposition 1 says that minimising (4) asymptotically recovers the target object for density sublevel set estimation, i.e., (1). Following Steinwart et al. [2005], we may also derive excess-risk bounds for $\ell_{01}^{(c)}$ in terms of $\ell^{(c)}$. This justifies reducing sublevel set estimation to classification of $P$ versus $\mu$.

**Density estimation as class-probability estimation**. We now observe that (4) can be used for estimating full densities as well. To do so, we move from classification-calibrated losses to the subset of *strictly proper composite* losses [Reid and Williamson, 2010]. These are the basic losses of *class-probability estimation* [Buja et al., 2005], and satisfy, for invertible *link* $\Psi\colon (0,1) \to \mathbb{R}$,

$$(\forall \theta \in (0,1)) \operatorname*{argmin}_{v \in \mathbb{R}} \underset{\mathsf{Y} \sim \mathrm{Bern}(\theta)}{\mathbb{E}} [\ell(\mathsf{Y}, v)] = \Psi(\theta). \tag{6}$$

In words, (6) stipulates that when using $\ell$ to distinguish positive and negative samples, it is optimal to predict (an invertible transformation of) the positive class-probability. A canonical example is the logistic loss $\ell(y, v) = \log(1 + e^{-yv})$ with link the logit function $\Psi\colon u \mapsto \log \frac{u}{1-u}$.

Applying strictly proper composite $\ell$ to distinguish $P$ from $\mu$ in (4), the Bayes-optimal solution is a transform of the underlying density. For $\ell$ being the logistic loss, this observation has been made previously to motivate reducing unsupervised to supervised learning [Hastie et al., 2009, Section 14.2.4]. The generalisation is trivial, but will prove to have useful further implications.

**Proposition 2:** *Let $\ell$ be a strictly proper composite loss with link $\Psi$. If $f^* \in \operatorname{Argmin}_f R(f; P, \mu, \ell)$, then for $\mu$-almost every $x \in \mathcal{X}$, $f^*(x) = \Psi_{\mathrm{rat}}(p(x))$, for the "ratio transformed" link function*

$$\Psi_{\mathrm{rat}}\colon z \mapsto \Psi\left(\frac{z}{1+z}\right). \tag{7}$$

---

**Example 3:** Consider the strictly proper composite loss

$$\ell(+1, v) = -v \qquad \ell(-1, v) = \frac{1}{2} \cdot v^2, \tag{8}$$

with link $\Psi(u) = u/(1-u)$, and corresponding $\Psi_{\mathrm{rat}}(z) = z$. Following Kanamori et al. [2009], we term this the "LSIF" loss. Minimising its risk performs least squares minimisation of $f$ versus $p$:

$$R(f) = \underset{\mathsf{X} \sim P}{\mathbb{E}}[-f(\mathsf{X})] + \frac{1}{2} \cdot \int_{\mathcal{X}} f(x)^2 \, \mathrm{d}\mu(x) = \frac{1}{2} \cdot \int_{\mathcal{X}} [f(x) - p(x)]^2 \, \mathrm{d}\mu(x) + \mathrm{const}. \tag{9}$$

It is thus Bayes-optimal to predict $f^* = p$, i.e., recover the underlying density. This fact has been noted in the context of density ratio estimation [Kanamori et al., 2009].

---

### 3.2 Towards calibrated anomaly detection

It is not hard to show that if $\ell$ is strictly proper composite, then so is the loss $\ell^{(c)}$ of (5) [Menon and Ong, 2016, Lemma 5]. Further, every such $\ell$ is also classification-calibrated [Reid and Williamson, 2010, Theorem 16]. Propositions 1 and 2 thus imply that if $f^*$ is the risk minimiser for $\ell^{(c)}$, computing $\mathrm{sign}(f^*)$ yields the target for density sublevel set estimation, while $f^*$ by itself yields the (transformed) target for density estimation. The hardness of density estimation discourages against the latter.[2] However, the fact that the risk $R(\cdot)$ accommodates two extremes of the problem space offers hope in using it to address the "intermediate" problem of calibrated anomaly detection per (2).

Tackling calibrated anomaly detection poses several challenges, however. We need to
  (i) obtain density $p$-values $F_\mathsf{P}(p(x))$, rather than raw density values $p(x)$,
 (ii) focus attention on density values lower than a threshold, and
(iii) (ideally) have the threshold correspond to a specified quantile $q \in (0, 1)$ of $p(\mathsf{X})$.

For (i), we may start with some base scorer $f\colon \mathcal{X} \to \mathbb{R}$, and compute a non-parametric estimate of the $p$-value via (3). For (ii), this scorer must preserve the order of the tail values of the density. This suggests we modify (2) and focus on the problem of *partial density estimation*, where given $\alpha > 0$, the goal is to produce a scorer $f\colon \mathcal{X} \to \mathbb{R}$ satisfying

$$f(x) \in \begin{cases} \{p(x)\} & \text{if } p(x) < \alpha \\ [\alpha, 1] & \text{if } p(x) > \alpha. \end{cases} \tag{10}$$

Density estimation (§3.2)     Partial density (§4.2)    Quantile control (§5)      Kernel absorption (§6)

$$\min_{f} \mathbb{E} - f(\mathsf{X}) + \tfrac{1}{2}\|f\|_{L_2}^2 \;\longrightarrow\; \min_{f,\alpha} \mathbb{E}\left[\alpha - f(\mathsf{X})\right]_+ + \tfrac{1}{2}\|f \wedge \alpha\|_{L_2}^2 - q\alpha \;\longrightarrow\; \min_{f,\alpha} \mathbb{E}\left[\alpha - f(\mathsf{X})\right]_+ + \tfrac{1}{2}\|f\|_{\mathcal{H}_\gamma}^2 - q\alpha$$

Figure 1: Summary of our approach for the LSIF loss from Example 3. Starting from a loss for full density estimation, we focus on only the tail of the densities by capping the loss (§4.2), add quantile control by adding a linear term related to the pinball loss (§5), and allow for tractable optimisation using a kernel absorption trick of Flaxman et al. [2017] (§6). Here, $\|f\|_{L_2}^2 = \int_{\mathcal{X}} f(x)^2 \, \mathrm{d}\mu(x)$.

Finally, for (iii), we need to somehow automatically relate $\alpha$ to the quantile $\alpha_q$.

We will now see how to suitably modify the loss in (4) to solve both (ii) and, at least implicitly, (iii); Figure 1 summarises. Our basic approach is to interpolate between the losses for density sublevel set and density estimation, noting that the former problem involves extracting a single sublevel set $\{x \colon p(x) < \alpha\}$ for given $\alpha > 0$, and the latter the *entire family* of sublevel sets $\bigcup_{\alpha'}\{x \colon p(x) < \alpha'\}$. We thus seek losses which are suitable for a *partial family* of sublevel sets $\bigcup_{\alpha' \leq \alpha}\{x \colon p(x) < \alpha'\}$. Quantile control is achieved by relating the result to the generalised pinball loss [Ehm et al., 2016].

## 4 Losses for partial density estimation

In order to obtain the optimal scorer for partial density estimation in (10), one could in theory simply perform full density estimation. However, this would entail solving a harder problem than we require. Can we focus attention on the *tail* of the density, so that no effort is placed on values larger than a threshold $\alpha > 0$? We will show how to do this using the *weight function* view of a proper loss.

### 4.1 Weight functions and density (sublevel set) estimation

Modelling only the tail of the density via (4) requires moving beyond strictly proper composite losses: from Proposition 2, *all* such losses have the *same* Bayes-optimal solution, up to a transformation. We will construct suitable alternate losses via the *weight function* representation of a proper composite loss: every such loss with invertible link $\Psi$ can be written [Reid and Williamson, 2010, Theorem 6]

$$\ell \colon (y, v) \mapsto \int_0^1 w(c) \cdot \ell_{01}^{(c)}(y, \Psi^{-1}(v)) \, \mathrm{d}c, \tag{11}$$

where the (generalised) function $w \colon [0, 1] \to \bar{\mathbb{R}}_{\geq 0}$ is the *weight*, and $\ell_{01}^{(c)}$ is the cost-sensitive loss

$$(\forall u \in [0, 1]) \, \ell_{01}^{(c)}(+1, u) \doteq (1 - c) \cdot [\![u < c]\!] \qquad \ell_{01}^{(c)}(-1, u) \doteq c \cdot [\![u > c]\!]. \tag{12}$$

Thus, minimising $\ell$ equivalently minimises a mixture of cost-sensitive losses for various cost ratios, whose relative importance is determined by $w$; see Buja et al. [2005] for several examples of such $w$.[3]

To get some intuition on the role of $w$, let us re-interpret our previous results through this lens. For $\alpha > 0$, pick $w(c) = \delta(c - c_\alpha)$, a Dirac delta-function centred at $c_\alpha = \alpha/(1 + \alpha)$. This weight corresponds (for a suitable link function) to the cost-sensitive loss $\ell_{01}^{(c_\alpha)}$ of (12), which by Proposition 1 estimates the $\alpha$th density sublevel set. Intuitively, by having $w$ be non-zero only at $c_\alpha$, we encourage modelling *only* the single sublevel set $\{x \in \mathcal{X} \mid p(x) < \alpha\}$.

If we instead pick $w(c) = (c \cdot (1 - c))^{-1}$, then (11) corresponds (for the logistic link function) to the logistic loss [Buja et al., 2005, Equation 19]. Proposition 2 shows this loss estimates the *entire* density. Intuitively, by having $w$ be non-zero everywhere, we encourage modelling of an entire *family* of sublevel sets $\{x \in \mathcal{X} \mid p(x) < \alpha\}$ for every $\alpha > 0$, which is equivalent to modelling the density.

Inspired by this, we now design suitable losses for partial density estimation.

### 4.2 Partial density estimation via $c_\alpha$-strictly proper losses

Suppose we only want to model density values below some $\alpha > 0$, per (10). A natural idea is to start with a loss having strictly positive weight function $w \colon [0, 1] \to \bar{\mathbb{R}}_{>0}$, which we modify to

$$\bar{w} \colon c \mapsto [\![c \leq c_\alpha]\!] \cdot w(c) \tag{13}$$

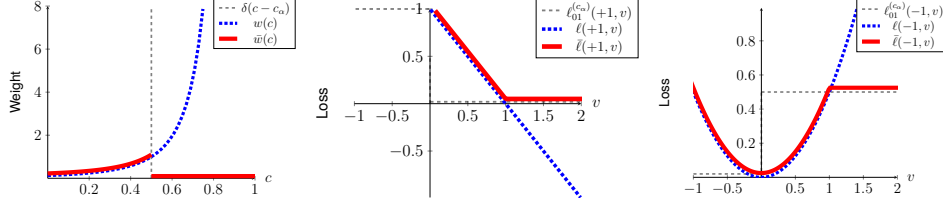

Figure 2: Illustration of weight functions and losses. Given cost threshold $c_\alpha = 0.5$, the weight $\delta(c - c_\alpha)$ gives a cost-sensitive loss suitable for density sublevel set estimation; $w(\cdot)$ gives a strictly proper loss suitable for density estimation; and $\bar{w}$ gives a $c_\alpha$-strictly proper loss suitable for partial density estimation. The losses corresponding to $\bar{w}$ are saturated versions of those corresponding to $w$.

for $c_\alpha = \alpha/(1 + \alpha)$, so that we place *no* weight on cost ratios above $c_\alpha$. Intuitively, we will pay no attention to modelling the sublevel sets for thresholds above $\alpha$, i.e., on modelling densities above $\alpha$. We will call the loss resulting from this $\bar{w}$ a $c_\alpha$-*strictly proper loss*.

Given a base loss $\ell$, we may explicitly compute the form of $\bar{\ell}$ induced from (13).

**Proposition 4:** *Pick any $\alpha > 0$ and strictly proper composite $\ell$ with link $\Psi \colon (0,1) \to \mathbb{R}$. Suppose $\bar{\ell}$ is the loss with weight function given by (13). Then, for $\rho_\alpha = \Psi_{\mathrm{rat}}(\alpha)$, up to constant translation,*

$$(\forall v \in \mathbb{R}) \; \bar{\ell}(+1, v) = \ell(+1, v \wedge \rho_\alpha) \qquad \bar{\ell}(-1, v) = \ell(-1, v \wedge \rho_\alpha). \tag{14}$$

*Further, if $f^* \in \operatorname{Argmin}_f R(f; P, \mu, \bar{\ell})$, then for $\mu$-almost every $x \in \mathcal{X}$,*

$$p(x) < \alpha \implies f^*(x) = \Psi_{\mathrm{rat}}(p(x)) \quad and \quad p(x) \geq \alpha \implies f^*(x) \geq \Psi_{\mathrm{rat}}(\alpha). \tag{15}$$

Figure 2 provides an illustrative example. Observe that if $\ell(+1, \cdot)$ is strictly decreasing, $\bar{\ell}(+1, v) = [\ell(+1, v) - \ell(+1, \rho_\alpha)]_+$ up to constant translation, explicating that the loss *saturates* beyond $\rho_\alpha$.

From (15), the new Bayes-optimal scorer will only model densities less than $\alpha$: for larger densities, any prediction larger than $\alpha$ suffices. This additional flexibility may prove useful in the pervasive scenario when one learns with a restricted class of scorers $\mathcal{F} \subset \mathbb{R}^{\mathcal{X}}$: by ignoring the behaviour of the density above $\alpha$, we may be able to select a scorer from $\mathcal{F}$ which better models the tail densities.

When $\alpha = \sup_x p(x)$, partial density estimation is identical to full density estimation, which is challenging. As $\alpha \to 0$, the estimation problem becomes less constrained and thus intuitively easier[4], with the limiting case $\alpha = 0$ trivially having every scorer as optimal. For fixed $\alpha$, the choice of underlying weight $w$ influences the precise range of density values where modelling effort is spent.

---

**Example 5:** For the LSIF loss in Example 3, one may show $w(c) = (1 - c)^{-3}$ [Menon and Ong, 2016, Table 1]. Unwrapping (14), the $c_\alpha$-strictly proper version of the loss is

$$\bar{\ell}(+1, v) = -(v \wedge \rho_\alpha) \qquad \bar{\ell}(-1, v) = {}^1\!/_2 \cdot (v \wedge \rho_\alpha)^2$$

with $\rho_\alpha = \alpha$. Upto translation, $\bar{\ell}(+1, \cdot) = [\rho_\alpha - v]_+$ is the hinge loss. The risk is (c.f. (9))

$$R(f) = \frac{1}{2} \cdot \int_{\mathcal{X}} (p(x) - (f(x) \wedge \rho_\alpha))^2 \, \mathrm{d}\mu(x) + \mathrm{const} \tag{16}$$

so that we perform least-squares minimisation with a *capped* version of our predictor. Evidently, the Bayes-optimal solution $f^*$ is exactly the partial density estimation target (10)

$$p(x) \leq \alpha \implies f^*(x) = p(x) \quad and \quad p(x) > \alpha \implies f^*(x) \geq \alpha.$$

---

In practice, one does not expect to be in a Bayes-optimal regime: thus, empirically minimising (16), e.g., will not yield exact tail densities. Nonetheless, the scores should approximately preserve the density ordering, owing to the relation between proper loss minimisation and ranking [Agarwal, 2013]; this suffices for the empirical $p$-value per (3). Note also that minimising the ranking risk with exponential surrogate equivalently minimises its classification counterpart [Ertekin and Rudin, 2011].

## 4.3 Partially capped losses

The loss $\bar{\ell}$ of (14) yields Bayes-optimal solutions suitable for density sublevel set estimation, but at price: in general, the loss will be non-convex. This is because a strictly proper composite loss has strictly increasing $\ell(-1, \cdot)$, meaning the capped version $\bar{\ell}(-1, \cdot)$ will saturate, and thus be non-convex. To resolve this, one may naïvely remove the saturation, yielding the *partially capped* loss

$$(\forall v \in \mathbb{R}) \, \tilde{\ell}(+1, v) = \ell(+1, v \wedge \rho_\alpha) \qquad \tilde{\ell}(-1, v) = \ell(-1, v). \tag{17}$$

Surprisingly, this seemingly simple-minded correction retains admissible Bayes-optimal solutions: one now obtains a saturated version of the underlying density. This has been previously observed for a specific $\ell$ in Vert and Vert [2006, Section 7.1].

**Proposition 6:** *Pick any $\alpha > 0$, and let $\tilde{\ell}$ be a partially capped loss as per (17). If $f^* \in$ $\mathrm{Argmin}_f R(f; P, \mu, \tilde{\ell})$, then for $\mu$-almost every $x \in \mathcal{X}$, $f^*(x) = \Psi_{\mathrm{rat}}(p(x) \wedge \alpha)$.*

Observe that when $p(x)$ is above $\alpha$, the Bayes-optimal scorers no longer have complete flexibility in their predictions: instead, they are clamped at exactly $\alpha$. Intuitively, the problem of estimating such capped densities is easier than full density estimation, but harder than partial density estimation. Thus, if non-convexity is not an issue, it may be preferable to simply use $c_\alpha$-strictly proper losses. On the other hand, we shall shortly see that partially proper losses can provide some quantile control.

---

**Example 7:** Unwrapping (17), the modified version of the LSIF loss in Example 3 has

$$\tilde{\ell}(+1, v) = [\rho_\alpha - v]_+ \qquad \tilde{\ell}(-1, v) = \nicefrac{1}{2} \cdot v^2, \tag{18}$$

where again $\rho_\alpha = \alpha$. Observe that $\tilde{\ell}(-1, \cdot)$ is not capped. The risk is (c.f. (16))

$$R(f) = \mathop{\mathbb{E}}_{\mathsf{X} \sim P} [\rho_\alpha - f(\mathsf{X})]_+ + \frac{1}{2} \cdot \int_{\mathcal{X}} f(x)^2 \, \mathrm{d}\mu(x) \tag{19}$$

$$= \frac{1}{2} \cdot \int_{\mathcal{X}} \left\{ [\![ f(x) \leq \rho_\alpha ]\!] \cdot (p(x) - f(x))^2 + [\![ f(x) \geq \rho_\alpha ]\!] \cdot f(x)^2 \right\} \mathrm{d}\mu(x) + \mathrm{const},$$

so that we perform least-squares minimisation with a *capped* version of our predictor, with an additional penalisation term. The Bayes-optimal solution is the capped density $f^*(x) = p(x) \wedge \alpha$.

---

# 5 Quantile control via the pinball loss

We consider the last challenge highlighted in §3.2: achieving quantile control. Concretely, given a desired quantile level $q \in (0, 1)$, how can we automatically infer (a bound on) the suitable density threshold $\alpha_q$? A natural idea is to incorporate the *pinball loss* [Steinwart and Christmann, 2008, Equation 2.26] into our objective. Recall that for $q \in (0, 1)$, this is the asymmetric linear loss $\phi(z; q) \doteq (1 - q) \cdot [z]_+ + q \cdot [-z]_+$. The distributional minimiser of this loss produces the $q$th quantile of a distribution [Steinwart and Christmann, 2008, Proposition 3.9]: for a real-valued $\mathsf{F} \sim F$, any $\rho^* \in \mathrm{argmin}_{\rho \in \mathbb{R}} \mathbb{E}\left[\phi(\rho - \mathsf{F}; q)\right]$ is the $q$th quantile of $F$.

One would like to apply this loss to the distribution of our scorer's predictions, i.e., the distribution of $\mathsf{F} \doteq f(\mathsf{X})$, where $\mathsf{X} \sim P$. Naïvely, one might think to first estimate a scorer $f$, and to then find the quantile. However, one cannot do this in conjunction with modelling only the density tail: to even construct a $c_\alpha$-strictly proper (or partially proper) loss, we require prior specification of the density threshold $\alpha > 0$, which by Proposition 4 maps to a threshold $\rho_\alpha = \Psi(\frac{\alpha}{1+\alpha})$ on the scores.

Fortunately, it is simple to jointly learn the scorer $f$ and threshold $\rho_{\alpha_q}$ with partially proper losses. The key is the following elementary observation, relating the pinball and capped linear loss.

**Lemma 8:** *For any $q \in (0, 1)$ and $z \in \mathbb{R}$, $\phi(z; q) = [z]_+ - q \cdot z$.*

To see why this is useful, observe that minimising the pinball loss on scores $\mathsf{F} = f(\mathsf{X})$ is equivalently

$$\min_\rho \mathop{\mathbb{E}}_{\mathsf{X} \sim P} \left[\phi\left(\rho - f(\mathsf{X}); q\right)\right] = \min_\rho \left[ \mathop{\mathbb{E}}_{\mathsf{X} \sim P} [\rho - f(\mathsf{X})]_+ - q \cdot \rho \right] + q \cdot \mathop{\mathbb{E}}_{\mathsf{X} \sim P} [f(\mathsf{X})].$$

The second term on the right hand side is independent of $\rho$, and may be ignored. The first term has the partially proper loss $\tilde{\ell}(+1, \cdot)$ of (18) with parameter $\rho$, and an additional term $(-q \cdot \rho)$ independent of $f$. Thus, if we add this term to our risk, it does not affect the optimisation over $f$; however, if we now optimise over $\rho$ as well, we will get a quantile for $f(\mathsf{X})$.

This idea can be generalised to any partially capped loss $\tilde{\ell}$ (17) that builds on some strictly proper composite $\ell$.[5] Concretely, we take such a loss with parameter $\rho$ and construct the modified risk

$$
\begin{aligned}
R_q(f, \rho; \ell) &\doteq R(f; \tilde{\ell}) + q \cdot \ell(+1, \rho) \\
&= \underset{\mathsf{X} \sim P}{\mathbb{E}}[\ell(+1, f(\mathsf{X})) - \ell(+1, \rho)]_+ + \int_{\mathcal{X}} \ell(-1, f(\mathsf{X})) \, \mathrm{d}\mu(x) + q \cdot \ell(+1, \rho),
\end{aligned}
\tag{20}
$$

which we now optimise over both the scorer $f$ and threshold $\rho$. The following guarantee relies on relating the first and third terms to the generalised pinball loss functions for quantile estimation [Ehm et al., 2016, Equation 5]; effectively, one just transforms the inputs to the pinball loss through $\ell(+1, \cdot)$.

**Proposition 9:** *Pick any $q \in (0, 1)$ and strictly proper composite loss $\ell$. If $(f^*, \rho^*) \in \mathrm{Argmin}_{f, \rho} R_q(f, \rho; q)$ then $\rho^*$ is the $q$th quantile of $f^*(\mathsf{X})$, for $\mathsf{X} \sim P$.*

A subtlety with the above is that one obtains the $q$th quantile of $f^*(\mathsf{X})$; however, this is *not* the same as obtaining the $q$th quantile of $p(\mathsf{X})$. The reason is simple: the optimal $f^*(\mathsf{X})$ is itself a capped version of $p(\mathsf{X})$, and so the quantiles of the two quantities will not coincide. Nonetheless, we are guaranteed to get a quantity that is *bounded* by the $q$th quantile of $p(\mathsf{X})$, and the above parametrisation thus offers an intuitive control knob; see Appendix C for more discussion.

---

**Example 10:** For the LSIF loss in Example 3, the modified risk over $f$ and $\rho$ is (c.f. (19))

$$
R_q(f, \rho) = \underset{\mathsf{X} \sim P}{\mathbb{E}}[\rho - f(\mathsf{X})]_+ + \int_{\mathcal{X}} \frac{1}{2} \cdot f(x)^2 \, \mathrm{d}\mu(x) - q \cdot \rho.
\tag{21}
$$

---

**Remark 11:** We can equally use (20) for density sublevel set estimation: we can minimise (20) and construct $\mathrm{sign}(f^*(x))$. This will produce an estimate of the sublevel set where $p(x) < \rho^*$ (by Proposition 4), and $\rho^*$ corresponds to a quantile of $f^*$ (by Proposition 9).

---

## 6 Optimisation without integral quadrature

The risk $R_q$ of (20) asymptotically recovers capped density estimates, and provides implicit quantile control. But given samples $\{x_n\}_{n=1}^N \sim P$, how do we practically optimise $R_q$? A natural strategy is to replace the expectation over $P$ with its empirical counterpart, and minimise

$$
\hat{R}_q(f, \rho) = \frac{1}{N} \cdot \sum_{n=1}^N [\ell(+1, f(x_n)) - \ell(+1, \rho)]_+ + \int_{\mathcal{X}} \ell(-1, f(x)) \, \mathrm{d}\mu(x) + q \cdot \ell(+1, \rho).
\tag{22}
$$

While the first term is a standard empirical risk, the second term is more problematic: when $\mathcal{X}$ is high-dimensional, approximating the integral, e.g. via quadrature, will require a large number of samples. Fortunately, when using kernelised scorers, a simple trick lets us deal with this issue.

### 6.1 A kernel absorption trick

We now show how a specific choice of loss lets us avoid explicit integration. This uses a trick developed in the context of point processes [McCullagh and Møller, 2006, Flaxman et al., 2017, Walder and Bishop, 2017]. Suppose that $\mathcal{X}$ is compact with $\mu(\mathcal{X}) < +\infty$. Suppose we choose $f$ from an RKHS $\mathcal{H}$ with continuous kernel $k \colon \mathcal{X} \times \mathcal{X} \to \mathbb{R}$. For $\gamma > 0$, define the regularised empirical risk

$$
\hat{R}_q(f, \rho; \gamma) \doteq \hat{R}_q(f, \rho) + \frac{\gamma}{2} \cdot \|f\|_{\mathcal{H}}^2.
$$

Now suppose $\ell(-1, v) = 1/2 \cdot v^2$. Then, Flaxman et al. [2017] showed that by Mercer's theorem,

$$
\int_{\mathcal{X}} \frac{1}{2} \cdot f(x)^2 \, \mathrm{d}\mu(x) + \frac{\gamma}{2} \cdot \|f\|_{\mathcal{H}}^2 = \frac{1}{2} \cdot \|f\|_{\mathcal{H}(\gamma, \mu)}^2,
\tag{23}
$$

where $\bar{\mathcal{H}}(\gamma, \mu)$ is a *modified* RKHS, corresponding to the integral operator $T_{\bar{k}} \doteq T_k(T_k + \gamma I)^{-1}$. While somewhat abstract, one can explicitly compute the corresponding kernel $\bar{k}$ when the Mercer expansion of $k$ is known, and use numerical approximations otherwise [Flaxman et al., 2017, Section 4]. Consequently, for $\ell(-1, v) = 1/2 \cdot v^2$ and $f \in \mathcal{H}$, the regularised empirical risk is *equivalent* to

$$\hat{R}_q(f, \rho; \gamma) = \frac{1}{N} \sum_{n=1}^{N} [\ell(+1, f(x_n)) - \ell(+1, \rho)]_+ + q \cdot \ell(+1, \rho) + \frac{1}{2} \cdot \|f\|_{\bar{\mathcal{H}}(\gamma, \mu)}^2.$$

One may now appeal to the representer theorem, and perform optimisation without any quadrature.

> **Example 12:** For the LSIF loss in Example 3, the regularised risk with a kernelised $f$ is (c.f. (21))
>
> $$R_q(f, \rho; \gamma) = \mathop{\mathbb{E}}_{\mathsf{X} \sim P} [\rho - f(\mathsf{X})]_+ - q \cdot \rho + \frac{1}{2} \cdot \|f\|_{\bar{\mathcal{H}}(\gamma, \mu)}^2. \qquad (24)$$

One subtlety with the empirical minimiser $\hat{f}^*$ of (24) is that $\hat{f}^*(x)$ could be negative for some $x$; this is inadmissible if we wish to interpret $\hat{f}^*(x)$ as an anomaly score. Following Kanamori et al. [2009], one may post-process scores to enforce non-negativity. More generally, one may start with an $\ell$ whose link $\Psi$ has image the entire $\bar{\mathbb{R}}$ rather than $\bar{\mathbb{R}}_+$, e.g., the logistic loss.

> **Remark 13:** We can equally use the above trick to ease optimisation for density sublevel set estimation, when employing $\ell(-1, v) = 1/2 \cdot v^2$ or its cost-weighted version.

## 6.2 Relation to one-class SVMs

We may contrast our objective in (24) to the one-class SVM (OC-SVM) [Schölkopf et al., 2001]. Fix an RKHS $\mathcal{H}$, and let $q \in (0, 1)$ be given. The primal OC-SVM objective is

$$R_q(f, \rho) = \mathop{\mathbb{E}}_{\mathsf{X} \sim P} [\rho - f(\mathsf{X})]_+ - q \cdot \rho + \frac{q}{2} \cdot \|f\|_{\mathcal{H}}^2.$$

This objective is almost identical to (24). The differences are that (24) regularises in a modified Hilbert space $\bar{\mathcal{H}}(\gamma, \mu)$, and that the regularisation strength is not tied to $q$. In (24), the regulariser $\|f\|_{\bar{\mathcal{H}}(\gamma, \mu)}^2$ conceptually plays two roles: it acts as a penalty on model complexity (owing to the $\|f\|_{\mathcal{H}}^2$ ingredient), and as a background loss (owing to the $\int_{\mathcal{X}} f(x)^2 \, d\mu(x)$ ingredient). Since $\bar{\mathcal{H}}(\gamma, \mu)$ is transparently derived from $\mathcal{H}$ given $\gamma > 0$, one has complete control over the model complexity for a given choice of $q \in (0, 1)$. By contrast, in the OC-SVM, fixing $q$ also fixes the model complexity.

Interestingly, Vert and Vert [2006] also related the Hilbert space norm $\|f\|_{\mathcal{H}}^2$ to $\int_{\mathcal{X}} f(x)^2 \, d\mu(x)$, in the limit of zero bandwidth for a Gaussian kernel. This was then used to analyse the Bayes-optimal solutions for a limiting version of the OC-SVM. The trick from Flaxman et al. [2017] in the previous section shows how to more generally relate these two norms through $\bar{\mathcal{H}}(\gamma, \mu)$.

More broadly, our framework offers a different perspective on the individual ingredients in a OC-SVM style objective: the hinge loss arises from capping a linear loss $\ell(+1, \cdot)$, so as to compute tail density estimates; the term linear in $\rho$ arises from re-expressing the hinge loss in terms of the pinball loss; and the regulariser arises from absorbing a model penalty plus a squared loss on the background distribution. Compared to the usual derivation of the OC-SVM, our objective arises from an explicit binary discrimination task, i.e., of $P$ versus $\mu$. We emphasise also that (24) is merely one special case of our framework, where the base $\ell$ is the LSIF loss (8). One may obtain different objectives by modifying $\ell$: e.g., using the logistic loss, we obtain a form of "one-class logistic regression".

## 7 Concluding remarks

We presented a loss function framework for calibrated anomaly detection, built upon an extant classification framework for density level-set problem [Steinwart et al., 2005]. We extended this framework to a class of proper losses, incorporated quantile control, and discussed means of avoiding explicit quadrature for optimisation. The framework also produced a close relative of the one-class SVM as a special case, giving a different perspective on the individual components of this method.

While our focus has been mostly conceptual, some illustrative experiments are presented in Appendix D. More broadly, viewing calibrated anomaly detection as loss minimisation also lets one interpret the problem as *estimating an entropy*; see Appendix E for further discussion.

## Acknowledgements

This work was supported by the Australian Research Council. It was motivated in part from a discussion with Zahra Ghafoori, whom we thank.

## Footnotes

*Now at Google Research.

[2]A nonparametric density estimator will, in a minimax sense, require exponentially many samples to yield a good approximation to the underlying density [Stone, 1980]. However, thresholding potentially rough density estimates may still yield a reasonable sublevel set estimator. This is analogous to the plugin approach to binary classification, where one estimates the underlying class-probability and thresholds it to make a classifier. Under some assumptions, plugin approaches can achieve fast classification rates [Audibert and Tsybakov, 2007].

[3]One can also interpret $w$ as a prior about cost ratios we will be evaluated on; see Appendix B.

[4]The claim of "easiness" of the problem is informal; we do not claim that changing $\alpha$ affects the minimax rate of convergence. We note however that even for sublevel set estimation, without some smoothness assumptions on $p$, the minimax rate will have exponential dependence on the dimensionality [Tsybakov, 1997, Theorem 4].

[5]We cannot apply this to a $c_\alpha$-strictly proper loss, as here $\bar{\ell}(-1, \cdot)$ will also depend on $\rho$.

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
