[Supplementary Material]

# Supplementary material for "A loss framework for calibrated anomaly detection"

## A  Proofs

We first present proofs of some helper results, before presenting those of results in the main body.

### A.1  Helper results

In the body, we considered losses corresponding to weights restricted to be non-zero on an interval. To generalise this idea, suppose $w\colon [0,1] \to \bar{\mathbb{R}}_+$ is the weight for some proper loss. Now for some $c_\alpha \in (0,1)$, consider a weight function $\bar{w}\colon [0,1] \to \bar{\mathbb{R}}_+$ is of the form

$$\bar{w}\colon c \mapsto \begin{cases} w(c) & \text{if } c \le c_\alpha \\ a \cdot w(c_\alpha) & \text{else,} \end{cases} \tag{25}$$

for fixed $c \in (0,1)$, $a \in [0,1]$. Intuitively, as $a \to 0$, we place less emphasis on cost ratios larger than $c_\alpha$. We may explicitly compute the form of the corresponding proper loss $\bar{\ell}$.

**Lemma 14:** *Let* $\lambda\colon \{\pm 1\} \times [0,1] \to \bar{\mathbb{R}}_+$ *be a strictly proper loss with weight function* $w\colon [0,1] \to \bar{\mathbb{R}}_+$. *Pick any* $c_\alpha \in (0,1)$ *and* $a \in [0,1]$. *Let* $\bar{\lambda}$ *be the proper loss with weight function given by Equation 25. Then, for each* $u \in (0,1)$,

$$\begin{aligned} \bar{\lambda}(-1, u) &= a \cdot \lambda(-1, u) + (1 - a) \cdot \lambda(-1, u \wedge c_\alpha) \\ \bar{\lambda}(+1, u) &= a \cdot \lambda(+1, u) + (1 - a) \cdot (\lambda(+1, u \wedge c_\alpha) - \lambda(+1, c_\alpha)). \end{aligned} \tag{26}$$

One may easily verify that when $a = 1$, we get the original loss $\ell$.

*Proof of Lemma 14.* By Shuford's integral formula [Reid and Williamson, 2010, Theorem 6],

$$\begin{aligned} \bar{\lambda}(-1, u) &= \int_0^1 \bar{\lambda}^{\mathrm{CS}(c)}(-1, u) \cdot \bar{w}(c)\,\mathrm{d}c \\ &= \int_0^u c \cdot \bar{w}(c)\,\mathrm{d}c \\ &= \int_0^{u \wedge c_\alpha} c \cdot \bar{w}(c)\,\mathrm{d}c + \int_{u \wedge c_\alpha}^u c \cdot \bar{w}(c)\,\mathrm{d}c \\ &= \int_0^{u \wedge c_\alpha} c \cdot w(c)\,\mathrm{d}c + \int_{u \wedge c_\alpha}^u a \cdot c \cdot w(c)\,\mathrm{d}c \\ &= \lambda(-1, u \wedge c_\alpha) + a \cdot (\lambda(-1, u) - \lambda(-1, u \wedge c_\alpha)) \\ &= a \cdot \lambda(-1, u) + (1 - a) \cdot \lambda(-1, u \wedge c_\alpha) \end{aligned}$$

$$\begin{aligned} \bar{\lambda}(+1, u) &= \int_0^1 \bar{\lambda}^{\mathrm{CS}(c)}(+1, u) \cdot \bar{w}(c)\,\mathrm{d}c \\ &= \int_u^1 (1 - c) \cdot \bar{w}(c)\,\mathrm{d}c \\ &= \int_u^{u \vee c_\alpha} (1 - c) \cdot \bar{w}(c)\,\mathrm{d}c + \int_{u \vee c_\alpha}^1 (1 - c) \cdot \bar{w}(c)\,\mathrm{d}c \\ &= \int_u^{u \vee c_\alpha} (1 - c) \cdot w(c)\,\mathrm{d}c + a \cdot \int_{u \vee c_\alpha}^1 (1 - c) \cdot w(c)\,\mathrm{d}c \\ &= \lambda(+1, u) - \lambda(+1, u \vee c_\alpha) + a \cdot \lambda(+1, u \vee c_\alpha) \\ &= a \cdot \lambda(+1, u) + (1 - a) \cdot (\lambda(+1, u) - \lambda(+1, u \vee c_\alpha)) \\ &= a \cdot \lambda(+1, u) + (1 - a) \cdot (\lambda(+1, u \wedge c_\alpha) - \lambda(+1, c_\alpha)), \end{aligned}$$

where the last line is because $f(x \wedge y) + f(x \vee y) = f(x) + f(y)$.  $\square$

## A.2 Results in the body

*Proof of Proposition 1.* Since $p = \frac{\mathrm{d}P}{\mathrm{d}\mu}$, we may write

$$R(f; P, \mu, \ell) = \underset{\mathsf{X} \sim P}{\mathbb{E}} [\ell(+1, f(\mathsf{X}))] + \int_{\mathcal{X}} \ell(-1, f(x)) \,\mathrm{d}\mu(x)$$

$$= \int_{\mathcal{X}} [p(x) \cdot \ell(+1, f(\mathsf{X})) + \ell(-1, f(x))] \,\mathrm{d}\mu(x).$$

For $\mu$-almost every $x \in \mathcal{X}$, the Bayes-optimal scorer for loss $\ell^{(c_\alpha)}$ is

$$f^*(x) \in \underset{v}{\operatorname{Argmin}} \left[ p(x) \cdot \ell^{(c_\alpha)}(+1, v) + \ell^{(c_\alpha)}(-1, v) \right]$$

$$= \underset{v}{\operatorname{Argmin}} \left[ r(x) \cdot \ell^{(c_\alpha)}(+1, v) + (1 - r(x)) \cdot \ell^{(c_\alpha)}(-1, v) \right] \text{ for } r(x) \doteq \frac{p(x)}{1 + p(x)} > 0$$

$$= \underset{v}{\operatorname{Argmin}} \left[ (1 - c_\alpha) \cdot r(x) \cdot \ell(+1, v) + c_\alpha \cdot (1 - r(x)) \cdot \ell(-1, v) \right]$$

$$= 2 \cdot [\![ (1 - c_\alpha) \cdot r(x) > c_\alpha \cdot (1 - r(x)) ]\!] - 1 \text{ by classification-calibration of } \ell$$

$$= 2 \cdot [\![ r(x) > c_\alpha ]\!] - 1$$

$$= 2 \cdot \left[\!\!\left[ p(x) > \frac{c_\alpha}{1 - c_\alpha} \right]\!\!\right] - 1 \text{ by definition of } r(x)$$

$$= 2 \cdot [\![ p(x) > \alpha ]\!] - 1 \text{ by definition of } c_\alpha.$$

$\square$

*Proof of Proposition 2.* Following the proof of Proposition 1, for $\mu$-almost every $x \in \mathcal{X}$, the Bayes-optimal scorer is

$$f^*(x) \in \underset{v}{\operatorname{Argmin}} \left[ r(x) \cdot \ell(+1, v) + (1 - r(x)) \cdot \ell(-1, v) \right] \text{ for } r(x) \doteq \frac{p(x)}{1 + p(x)} > 0$$

$$= \Psi(r(x)) \text{ by definition of a strictly proper composite loss (6)}$$

$$= \Psi \left( \frac{p(x)}{1 + p(x)} \right) \text{ by definition of } r(x).$$

$\square$

*Proof of Proposition 4.* Applying Lemma 14 with $a = 0$, the proper loss corresponding to $\bar{w}$ is

$$\bar{\lambda}(-1, u) = \lambda(-1, u \wedge c_\alpha)$$
$$\bar{\lambda}(+1, u) = \lambda(+1, u \wedge c_\alpha) - \lambda(+1, c_\alpha).$$

Since $\bar{\ell} = \bar{\lambda} \circ \Psi^{-1}$ and $c_\alpha = \Psi(\rho_\alpha)$,

$$\bar{\ell}(-1, u) = \ell(-1, v \wedge \rho_\alpha)$$
$$\bar{\ell}(+1, u) = \ell(+1, v \wedge \rho_\alpha) - \ell(+1, \rho_\alpha).$$

Note that $\ell(+1, \rho_\alpha)$ is a constant, which plays no role in optimisation and thus may be safely ignored.

For any $\eta \in [0, 1]$, define the conditional risk $L_\eta(v) \doteq \eta \cdot \ell(+1, v) + (1 - \eta) \cdot \ell(-1, v)$. Following the proof of Proposition 1, for $\mu$-almost every $x \in \mathcal{X}$, the Bayes-optimal scorer is

$$f^*(x) \in \underset{v}{\operatorname{Argmin}} \left[ r(x) \cdot \bar{\ell}(+1, v) + (1 - r(x)) \cdot \bar{\ell}(-1, v) \right] \text{ for } r(x) \doteq \frac{p(x)}{1 + p(x)} > 0$$

$$= \underset{v}{\operatorname{Argmin}} \left[ r(x) \cdot (\ell(+1, v \wedge \rho_\alpha) - \ell(+1, \rho_\alpha)) + (1 - r(x)) \cdot \ell(-1, v \wedge \rho_\alpha) \right]$$

$$= \underset{v}{\operatorname{Argmin}} \left[ r(x) \cdot \ell(+1, v \wedge \rho_\alpha) + (1 - r(x)) \cdot \ell(-1, v \wedge \rho_\alpha) \right]$$

$$= \underset{v}{\operatorname{Argmin}} L_{r(x)}(v \wedge \rho_\alpha)$$

Figure 3: Illustration of conditional risk for original loss $\ell(+1, v) = -v$ and $\ell(-1, v) = \frac{1}{2} \cdot v^2$, and its $c_\alpha$-strictly proper version $\bar{\ell}(+1, v) = -(v \wedge \rho)$ and $\ell(-1, v) = \frac{1}{2} \cdot (v \wedge \rho)^2$. We choose $\alpha = 1$, which corresponds to $\rho = \Psi(\frac{\alpha}{1+\alpha}) = 1$ as well. In the left plot, we show the conditional risks for $\eta = 0.2$, for which $\Psi(\eta) < \rho$. Confirming the theory, the minimiser for $\bar{L}$ is exactly $\Psi(\eta) = \frac{1}{4}$. In the right plot, we show the conditional risks for $\eta = 0.8$, for which $\Psi(\eta) > \rho$. Confirming the theory, the minimiser for $\bar{L}$ is any value in $[\rho, +\infty)$.

$$= \operatorname*{Argmin}_{v} \bar{L}_{r(x)}(v),$$

where $\bar{L}_\eta(v) \doteq L_\eta(v \wedge \rho_\alpha)$.

For any strictly proper loss, $L_\eta$ is strictly quasi-convex for any $\eta \in [0, 1]$ [Reid and Williamson, 2010, Theorem 23]. Further, it has a unique minimum at $v^* = \Psi(\eta)$, by definition of strict properness. Consequently, $L_\eta$ must be strictly decreasing on $[0, v^*]$ and strictly increasing on $[v^*, 1]$. Observe that $\bar{L}_\eta$ and $L_\eta$ coincide on $[0, \rho_\alpha]$, while $\bar{L}_\eta$ remains constant on $[\rho_\alpha, +\infty)$. Thus, $\bar{L}_\eta$ must also be strictly decreasing on $[0, \rho_\alpha \wedge v^*]$, and constant on $[\rho_\alpha, +\infty)$.

Consider two cases for the minimiser of $\bar{L}_\eta$:

(a) suppose $v^* < \rho_\alpha$. Then, since $\bar{L}_\eta$ and $L_\eta$ coincide on $[0, \rho_\alpha]$, we have $\bar{L}_\eta(v^*) < \bar{L}_\eta(v)$ for any $v \in [0, \rho_\alpha] - \{v^*\}$. But we know $\bar{L}_\eta$ is constant on $[\rho_\alpha, +\infty)$, and so $\bar{L}_\eta(v^*) < \bar{L}_\eta(v)$ for any $v$ on this interval as well. Thus, the minimiser must occur at $v = v^*$.

(b) suppose $v^* \geq \rho_\alpha$. Then, we must have that $L_\eta$ and $\bar{L}_\eta$ are strictly decreasing on $[0, \rho_\alpha]$. This means that $\rho_\alpha$ is the minimiser of $\bar{L}_\eta$ on $[0, \rho_\alpha]$. Now, since $\bar{L}_\eta$ is constant on $[\rho_\alpha, +\infty)$, it follows that any point in this interval is also a minimiser.

Thus, the optimal solution is $\bar{v}^* = v^*$ when $v^* < \rho_\alpha$, and $\bar{v}^* \geq \rho_\alpha$ when $v^* \geq \rho_\alpha$. Since $v^* = \Psi(\eta) = \Psi(r(x)) = \Psi_{\mathrm{rat}}(p(x))$, the result follows.

As an example, we illustrate the behaviour of the conditional risk in Figure 3.

$\square$

*Proof of Proposition 6.* Following the proof of Proposition 4, for $\mu$-almost every $x \in \mathcal{X}$, the Bayes-optimal scorer is

$$f^*(x) \in \operatorname*{Argmin}_{v} \left[ r(x) \cdot \bar{\ell}(+1, v) + (1 - r(x)) \cdot \bar{\ell}(-1, v) \right] \text{ for } r(x) \doteq \frac{p(x)}{1 + p(x)} > 0$$

$$\in \operatorname*{Argmin}_{v} \left[ r(x) \cdot \ell(+1, v \wedge \rho_\alpha) + (1 - r(x)) \cdot \ell(-1, v) \right]$$

$$\in \operatorname*{Argmin}_{v} \bar{L}_\eta(v),$$

where $\eta \doteq r(x)$ and $\bar{L}_\eta(v) \doteq \eta \cdot \ell(+1, v \wedge \rho_\alpha) + (1 - \eta) \cdot \ell(-1, v)$.

Recall that the conditional risk for $\ell$ is the strictly quasi-convex quantity $L_\eta: v \mapsto \eta \cdot \ell(+1, v) + (1 - \eta) \cdot \ell(-1, v)$. Further, it has a unique minimum at $v^* = \Psi(\eta)$, by definition of strict properness. We thus have that $L_\eta$ is strictly decreasing on $[0, v^*]$ and strictly increasing on $[v^*, 1]$.

We have $\bar{L}_\eta(v) = L_\eta(v)$ for $v \le \rho_\alpha$. For $v > \rho_\alpha$, we saturate the effect of the second term. Consider two cases for the minimiser of $\bar{L}_\eta$:

(a) suppose $v^* \le \rho_\alpha$. Then, since $v^*$ is the minimiser of $L_\eta$, and $\bar{L}_\eta(v) = L_\eta(v)$ on $[0, \rho_\alpha]$, it must also be the minimiser of $\bar{L}_\eta$ on $[0, \rho_\alpha]$. In particular, $\bar{L}_\eta(v^*) \le \bar{L}_\eta(\rho_\alpha)$. Further, the partial loss $\ell(-1, \cdot)$ is strictly increasing for any strictly proper composite loss; thus, $\bar{L}_\eta$ must be strictly increasing on $[\rho_\alpha, +\infty)$. Consequently, the minimiser must be $v^*$.

(b) suppose $v^* > \rho_\alpha$. Then, $L_\eta$ as well as $\bar{L}_\eta$ must be strictly decreasing on $[0, \rho_\alpha]$. Per the previous case, $\bar{L}_\eta$ must be strictly increasing on $[\rho_\alpha, +\infty)$. Consequently, the minimiser must be $\rho_\alpha$.

The minimiser is thus $v^* \wedge \rho_\alpha$. Since $v^* = \Psi(\eta) = \Psi(r(x)) = \Psi_{\mathrm{rat}}(p(x))$, the result follows.

As an example, we illustrate the behaviour of the conditional risk in Figure 4. Compared to Figure 4, observe that in the right plot, we do *not* have saturation of the modified conditional risk $\bar{L}_\eta$. As a result, the optimal solution is capped at $\rho_\alpha$, regardless of the value of $v^*$. $\qquad\square$

*Proof of Lemma 8.* We have

$$
\begin{aligned}
[z]_+ &= \begin{cases} z & \text{if } z \ge 0 \\ 0 & \text{else} \end{cases} \\
&= \begin{cases} (1-q) \cdot z + q \cdot z & \text{if } z \ge 0 \\ 0 & \text{else} \end{cases} \\
&= \begin{cases} (1-q) \cdot z & \text{if } z \ge 0 \\ -q \cdot z & \text{else} \end{cases} + q \cdot z \\
&= \phi(z; q) + q \cdot z.
\end{aligned}
$$

$\qquad\square$

*Proof of Proposition 9.* For any strictly decreasing function $h \colon \mathbb{R} \to \mathbb{R}$, and any $q \in (0, 1)$,

$$
\begin{aligned}
h(z \wedge \rho) - h(\rho) &= [h(z) - h(\rho)]_+ \\
&= \phi(h(z) - h(\rho); q) + q \cdot (h(z) - h(\rho)) \text{ by Lemma 8} \\
&= \phi(h(z) - h(\rho); q) + q \cdot h(z) - q \cdot h(\rho).
\end{aligned}
$$

For $\alpha > 0$ and strictly decreasing $\ell(+1, \cdot)$, let $\tilde{\ell}(+1, \cdot)$ be the partially proper loss with $\rho \doteq \rho_\alpha = \Psi(c_\alpha)$. Consider the modified risk

$R_q(f, \rho) \doteq R(f) + q \cdot \ell(+1, \rho)$

$$
\begin{aligned}
&= \operatorname*{\mathbb{E}}_{\mathsf{X} \sim P} [\ell(+1, f(\mathsf{X}) \wedge \rho) - \ell(+1, f(\mathsf{X}))] + q \cdot \ell(+1, \rho) + \int_\mathcal{X} \ell(-1, f(x)) \, \mathrm{d}\mu(x) \\
&= \operatorname*{\mathbb{E}}_{\mathsf{X} \sim P} [\ell(+1, f(\mathsf{X})) - \ell(+1, \rho)]_+ + q \cdot \ell(+1, \rho) + \int_\mathcal{X} \ell(-1, f(x)) \, \mathrm{d}\mu(x) \\
&= \operatorname*{\mathbb{E}}_{\mathsf{X} \sim P} [\phi(\ell(+1, f(\mathsf{X})) - \ell(+1, \rho); q)] + q \cdot \operatorname*{\mathbb{E}}_{\mathsf{X} \sim P} [\ell(+1, f(\mathsf{X}))] + \int_\mathcal{X} \ell(-1, f(x)) \, \mathrm{d}\mu(x),
\end{aligned}
$$

which we now optimise over both $f$ and $\rho$. For fixed $\rho$, the minimisation over $f$ is unchanged; for fixed $f$, the second and third terms are irrelevant, and so the minimisation over $\rho$ is

$$
\min_\rho \operatorname*{\mathbb{E}}_{\mathsf{X} \sim P} [\phi(\ell(+1, f(\mathsf{X})) - \ell(+1, \rho); q)]. \tag{27}
$$

The resulting minimiser will again be the $q$th quantile of the distribution of scores. This is owing to the calibration property of a generalisation of the pinball loss [Steinwart et al., 2014, Equation 15], [Ehm et al., 2016, Equation 5]: for strictly increasing $g \colon \mathbb{R} \to \mathbb{R}$, let

$$
\begin{aligned}
\bar{\phi}(\hat{y} - y; g, q) &\doteq |\, [\![ y \le \hat{y} ]\!] - q| \cdot |g(\hat{y}) - g(y)| \\
&= \begin{cases} (1-q) \cdot (g(\hat{y}) - g(y)) & \text{if } y \le \hat{y} \\ q \cdot (g(y) - g(\hat{y})) & \text{else} \end{cases} \\
&= \phi(g(\hat{y}) - g(y), q).
\end{aligned} \tag{28}
$$

Figure 4: Illustration of conditional risk for original loss $\ell(+1, v) = -v$ and $\ell(-1, v) = \frac{1}{2} \cdot v^2$, and its "partially saturated" version $\bar{\ell}(+1, v) = -(v \wedge \rho)$ and $\ell(-1, v) = \frac{1}{2} \cdot v^2$. We choose $\alpha = 1$, which corresponds to $\rho = \Psi(\frac{\alpha}{1+\alpha}) = 1$ as well. In the left plot, we show the conditional risks for $\eta = 0.2$, for which $\Psi(\eta) < \rho$. Confirming the theory, the minimiser for $\bar{L}$ is exactly $\Psi(\eta) = \frac{1}{4}$. In the right plot, we show the conditional risks for $\eta = 0.5$, for which $\Psi(\eta) = \rho$. Confirming the theory, the minimiser for $\bar{L}$ is exactly $\rho$.

As before, if for a distribution $F$ over $\mathbb{R}$, we pick

$$\rho^* \in \underset{\rho \in \mathbb{R}}{\text{Argmin}} \; \underset{\mathsf{F} \sim F}{\mathbb{E}} \left[ \bar{\phi}(\rho - \mathsf{F}; g, q) \right],$$

then $\rho^*$ is the $q$th quantile of $F$. For our problem, we simply need to set $g(v) = -\ell(+1, v)$ (recalling that $\ell(+1, \cdot)$ is strictly decreasing) to conclude that Equation 27 produces the $q$th quantile of $\mathsf{F} \doteq \hat{f}^*(\mathsf{X})$.

Let us remark here that the optimisation for $\rho$ is always quasi-convex, but will only be convex for $g(z) = z$, i.e., for $\ell(+1, v) = -v$ [Steinwart et al., 2014, Corollary 11]. However, one could define $\rho' = g(\rho)$ and optimise over $\rho'$ instead, taking care to compute $g^{-1}(\rho')$ for the final prediction. $\quad \square$

# B  Interpreting the weight function as a prior

Given infinite data, and an arbitrarily powerful function class, there is no reason to favour one strictly proper loss over another: every such loss results in the same Bayes-optimal solution, namely, a transform of the underlying class-probability. Neither of these assumptions holds in practice, of course. In this case, minimising different losses will result in different solutions. How then might one choose amongst different losses?

To obtain one possible answer, consider the following scenario. Suppose there is a distribution $D$ over $\mathcal{X} \times \{\pm 1\}$, representing a binary classification problem. One wishes to design a good classifier for this problem, as evaluated by a cost-sensitive loss $\ell_{01}^{(c)}$ for some $c \in (0, 1)$. *However*, there is uncertainty as to what this cost ratio will be: one must first design a model, and only then get informed as to what the cost-ratio is.

It is natural to express this uncertainty over cost-ratios in terms of a (possibly improper) prior distribution $\pi$ over $[0, 1]$. In this case, the natural strategy for the learner is to minimise the *expected cost-sensitive loss* under costs drawn from $\pi$. Concretely, we would find a scorer $f \colon \mathcal{X} \to \bar{\mathbb{R}}$ and link $\Psi \colon \bar{\mathbb{R}} \to [0, 1]$ to minimise

$$\mathop{\mathbb{E}}_{(\mathsf{X},\mathsf{Y}) \sim D} \left[ \mathop{\mathbb{E}}_{\mathsf{C} \sim \pi} \left[ \ell_{01}^{(\mathsf{C})}(\mathsf{Y}, \Psi^{-1}(f(\mathsf{X}))) \right] \right] = \mathop{\mathbb{E}}_{(\mathsf{X},\mathsf{Y}) \sim D} \left[ \ell(\mathsf{Y}, f(\mathsf{X})) \right],$$

where

$$\ell(y, v) = \int_0^1 \ell_{01}^{(c)}(y, \Psi^{-1}(v)) \cdot \pi(c) \, \mathrm{d}c.$$

From (11), we identify this as a proper composite loss corresponding to weight function $\pi$. Thus, this suggests that we should choose our weight function to reflect our prior belief as to which cost-ratios we expect to be evaluated on. (Observe that, trivially, if one's true belief were $\pi$, it would not be rational to use a loss corresponding to $\pi' \neq \pi$; this would necessarily yield a solution that is sub-optimal loss according to cost-ratios drawn from our belief $\pi$.)

A related idea is to tune the weight function based on cost-ratios of interest, e.g., [Buja et al., 2005, Section 14], [Hand and Vinciotti, 2003].

In the context of anomaly detection, recall from the proof of Proposition 9 that our choice of loss $\ell(+1, \cdot)$ relates to a particular choice of increasing function $g(\cdot)$ parametrising a generalised quantile elicitation loss (28). Such losses also have an integral representation [Ehm et al., 2016, Theorem 1a],

$$\bar{\ell}_{1-q}^{\mathrm{pin}}(y, \hat{y}; g) = \int_{-\infty}^{+\infty} \ell_{1-q}^{\mathrm{prim}}(y, \hat{y}; v) \cdot h(v) \, \mathrm{d}v,$$

where $h(v) = g'(v)$, and the "primitive" quantile loss is

$$\ell_{1-q}^{\mathrm{prim}}(y, \hat{y}; v) = (\llbracket y < \hat{y} \rrbracket - q) \cdot (\llbracket v < \hat{y} \rrbracket - \llbracket v < y \rrbracket).$$

These primitive losses are analogues of the cost-sensitive loss, and measure whether the prediction and ground truth are on the same side of a fixed threshold of $v$. For a strictly proper composite $\ell$ with weight $\pi$, we have [Reid and Williamson, 2010, Theorem 10]

$$-\ell'(+1, v) = \frac{\Psi^{-1}(v) - 1}{\Psi'(\Psi^{-1}(v))} \cdot \pi(\Psi^{-1}(v)).$$

Now note that the weighting over thresholds is $h(v) = g'(v) = -\ell'(+1, v)$. Consequently, the weighting $\pi$ over costs $c$ implicitly translates to a weighting $h$ over thresholds $v$. We may thus pick $\pi$ and $\Psi$ to reflect the portion of the distribution $\mathsf{F}$ we wish to focus attention on.

In turn, the choice of this function $g(\cdot)$ relates to the distribution of the random variable whose quantile we are eliciting. Concretely, consider the problem

$$\min_\rho \mathop{\mathbb{E}}_{\mathsf{F} \sim F} \left[ \phi_{1-q}(g(\rho) - g(\mathsf{F})) \right],$$

whose minimising argument returns the $q$th quantile of $\mathsf{F}$. Given access to the distribution $F$ as above, every choice of $g$ produces the same result. However, given an empirical approximation $\hat{F}$, this is no longer true.

# C  On the quantile-control parameter

In §5, we proposed a pinball-loss based approach to estimate a suitable density threshold $\alpha$ for a given control parameter $q \in (0, 1)$. We expand upon the precise nature of the optimal $\alpha^*$, explicating that while it will *not* be the $q$th quantile of $p(\mathsf{X})$ in general, we nonetheless get some meaningful control on the fraction of instances classified as anomalous.

Recall that for the underlying loss $\ell$ of Example 3, and parameter $q \in (0, 1)$, our "quantile-controlled" objective is to jointly optimise over a scorer $f$ and threshold $\alpha$ to minimise

$$R(f, \alpha) \doteq \underset{\mathsf{X} \sim P}{\mathbb{E}} [\alpha - f(\mathsf{X})]_+ + \frac{1}{2} \cdot \underset{\mathsf{X} \sim Q}{\mathbb{E}} \left[ \frac{1}{2} \cdot f(\mathsf{X})^2 \right] - q \cdot \alpha,$$

where for clarity, we use $\alpha$ instead of $\rho$ (since this loss has $\rho_\alpha = \alpha$), and assume $\mu(\mathcal{X}) = 1$ so that it corresponds to some probability distribution $Q$.

**Explicit quantile control: negative result** In Proposition 9, we argued that the optimal $\alpha^*$ will be the $q$th quantile of $f^*(\mathsf{X})$, for $\mathsf{X} \sim P$, since the objective can be related to the pinball loss. However, this does *not* mean that $\alpha^*$ will be the $q$th quantile of $p(\mathsf{X})$. To see this, recall from Proposition 6 that the optimal scorer $f^*(x) = p(x) \wedge \alpha^*$. Thus, by definition of a quantile,

$$P(p(\mathsf{X}) \wedge \alpha^* < \alpha^*) \leq q \leq P(p(\mathsf{X}) \wedge \alpha^* \leq \alpha^*).$$

This simplifies to

$$P(p(\mathsf{X}) < \alpha^*) \leq q \leq 1.$$

When $p(\mathsf{X})$ has continuous, invertible cumulative distribution function $F$, this implies $\alpha^* \leq F^{-1}(q)$. Consequently, $\alpha^*$ is merely a *lower bound* on the $q$th quantile of $p(\mathsf{X})$.

**Implicit quantile control: positive result**. While lacking explicit quantile control, our objective nonetheless provides *implicit* quantile control in the following sense: sweeping over the entire range of $q$ is equivalent to sweeping over the entire range of $\alpha^*$ values. But for fixed $q$, the corresponding $\alpha^*$ does not correspond to the $q$th quantile for the density.

To understand better the role of changing $q$, let us restrict attention to $f$ of the form $f(x) = p(x) \wedge \alpha$, and consider a univariate optimisation over $\alpha$ alone. (Clearly, this choice of $f$ will result in the same optimal solution for $\alpha$.) Now consider the objective with respect to $\alpha$ alone:

$$R(\alpha) \doteq \underset{\mathsf{X} \sim P}{\mathbb{E}} [\alpha - (p(\mathsf{X}) \wedge \alpha)]_+ + \frac{1}{2} \cdot \underset{\mathsf{X} \sim Q}{\mathbb{E}} \left[ \frac{1}{2} \cdot (p(\mathsf{X}) \wedge \alpha)^2 \right] - q \cdot \alpha$$

$$= \underset{\mathsf{X} \sim Q}{\mathbb{E}} \left[ p(\mathsf{X}) \cdot [\alpha - p(\mathsf{X})]_+ + \frac{1}{2} \cdot (p(\mathsf{X}) \wedge \alpha)^2 \right] - q \cdot \alpha.$$

Observe now that for any $p \geq 0$,

$$g(\alpha; p) \doteq p \cdot [\alpha - p]_+ + \frac{1}{2} \cdot (p \wedge \alpha)^2$$

$$= \begin{cases} \frac{1}{2} \cdot \alpha^2 & \text{if } \alpha \leq p \\ \alpha \cdot p - \frac{1}{2} \cdot p^2 & \text{else} \end{cases}$$

$$= \begin{cases} \frac{1}{2} \cdot (\alpha - p)^2 & \text{if } \alpha \leq p \\ 0 & \text{else} \end{cases} + \alpha \cdot p - \frac{1}{2} \cdot p^2$$

$$= \frac{1}{2} \cdot ([p - \alpha]_+)^2 + \alpha \cdot p - \frac{1}{2} \cdot p^2.$$

This function is evidently convex and differentiable with respect to $\alpha$. Since $R$ is constructed by integrating $g(\alpha; p)$ over $p$, it must also be convex and differentiable. More explicitly,

$$R(\alpha) = \underset{\mathsf{X} \sim Q}{\mathbb{E}} \frac{1}{2} \cdot ([p(\mathsf{X}) - \alpha]_+)^2 + (1 - q) \cdot \alpha + \text{constant}.$$

Consequently, the first-order gradient condition on $R$ implies that at optimality,

$$R'(\alpha^*) = 0 = \underset{\mathsf{X} \sim Q}{\mathbb{E}} - [p(\mathsf{X}) - \alpha^*]_+ + 1 - q.$$

Consequently, for $G(\alpha) \doteq \underset{\mathsf{X} \sim Q}{\mathbb{E}} [p(\mathsf{X}) - \alpha]_+$, we have

$$G(\alpha^*) = 1 - q.$$

Now, the function $G \colon \mathbb{R}_+ \to \mathbb{R}_+$ is an integral over $x$ of a family of functions that are continuous and strictly decreasing on $[0, p(x)]$. Consequently, $G$ is continuous and strictly decreasing on $[0, F^{-1}(1))$, where $F$ is the cumulative distribution function of $p(\mathsf{X})$. Further, at the endpoints we have $G(0) = 1$ and $G(F^{-1}(1)) = 0$.

As a result, there exists a continuous inverse $G^{-1}$ such that for any $q \in [0, 1]$, we have $\alpha^* = G^{-1}(1 - q)$. It follows that $\alpha^*$ varies continuously as $q$ is varied, and vice-versa. Further, when $q = 0$, $\alpha^* = 0$, while when $q \to 1$, $\alpha^* \to F^{-1}(1)$. Thus, at the boundaries, $\alpha^*$ coincides with the ordinary quantile of $p(\mathsf{X})$.

---

**Example 15:** To make the above concrete, consider the case where $P$ has density $p(x) = 2 \cdot x$ with respect to the uniform measure over $\mathcal{X} = [0, 1]$. The objective of interest is

$$
\begin{aligned}
R(\alpha) &= \underset{\mathsf{X} \sim Q}{\mathbb{E}} \left[ p(\mathsf{X}) \cdot [\alpha - p(\mathsf{X})]_+ + \frac{1}{2} \cdot (p(\mathsf{X}) \wedge \alpha)^2 \right] - q \cdot \alpha \\
&= \underset{\mathsf{X} \sim Q}{\mathbb{E}} \left[ 2 \cdot \mathsf{X} \cdot [\alpha - 2 \cdot \mathsf{X}]_+ + \frac{1}{2} \cdot (2 \cdot \mathsf{X} \wedge \alpha)^2 \right] - q \cdot \alpha \\
&= \int_0^1 2 \cdot x \cdot [\alpha - 2 \cdot x]_+ + \frac{1}{2} \cdot (2 \cdot x \wedge \alpha)^2 \, \mathrm{d}x - q \cdot \alpha \\
&= \frac{\alpha^3}{12} + \frac{\alpha^2}{2} - \frac{\alpha^3}{6} - q \cdot \alpha \text{ when } \alpha \in [0, 2] \\
&= -\frac{\alpha^3}{12} + \frac{\alpha^2}{2} - q \cdot \alpha \text{ when } \alpha \in [0, 2].
\end{aligned}
$$

This function is convex, with

$$R'(\alpha) = -\frac{\alpha^2}{4} + \alpha - q.$$

Thus, the optimal choice of $\alpha$ is

$$\alpha^* = 2 \cdot (1 - \sqrt{1 - q}).$$

Clearly, $\alpha^*$ varies from 0 to 2 as $q$ is varied from 0 to 1. We can further explicitly compute the quantile of $p(\mathsf{X})$ corresponding to this choice of $\alpha^*$ as

$$P(p(\mathsf{X}) \le \alpha^*) = P(\mathsf{X} \le \alpha^*/2) = \left( \frac{\alpha^*}{2} \right)^2 = 2 - q - 2 \cdot \sqrt{1 - q}.$$

That is, as $q$ is varied, we capture in a nonlinear manner a suitable quantile of $p(\mathsf{X})$.

---

# D Experimental illustration

We present experiments illustrating the behaviour of methods introduced in the body of the paper.

## D.1 Visualisation of optimal scorers

For our first experiment, we visualise the optimal scorers for various methods in a simple one-dimensional example. Specifically, we consider the Bayes-optimal solutions for:

- full density estimation, minimising the LSIF loss $\ell$ of Example 3
- partial density estimation with the "partially capped" loss $\tilde{\ell}$ of Example 7

### D.1.1 Experimental setup

We consider a 1D distribution on $\mathcal{X} = [0, 1]$, with density $p(x) = 2 \cdot x$ with respect to the Lebesgue measure. We draw $n = 10^4$ samples from the distribution. For a given quantile level $q \in (0, 1)$, we minimise the empirical, quantile-corrected risk using Gaussian-kernelised scorers. Since the Gaussian kernel is univeral, the optimal solution is expected to mimic the Bayes-optimal solution. The bandwidth of the kernel was chosen by cross-validation, so as to optimise the log-likelihood of standard kernel density estimation. For ease of optimisation, we use the Nyström approximation to the kernel with $k = 500$ prototypes. For simplicity, we do not employ the kernel absorption trick; rather, we draw $n$ samples from the uniform distribution over $[0, 1]$ to approximate the $\int_{\mathcal{X}} \tilde{\ell}(-1, f(x)) \, \mathrm{d}\mu(x)$ term.

### D.1.2 Experimental results

Figure 5 shows the optimal kernelised solution. We observe that, as predicted by the theory:

- full density estimation essentially recovers the underlying $p$
- partially capped loss minimisation recovers a capped version of the density

## D.2 Calibration performance on synthetic data

In our next experiment, we validate that minimising a partially proper loss yields good density tail estimates, and that on finite samples these estimates tend to be superior to those produced by full density estimation.

To control all sources of error, we perform experiments on a number of synthetically generated distributions:

- `linear`, a distribution on $[0, 1]$ with density $p(x) = 2 \cdot x$.
- `arcsin`, an arcsine distribution on $[0, 1]$ with density $p(x) = (\pi \cdot \sqrt{x \cdot (1 - x)})^{-1}$.
- `truncnorm`, a truncated standard Gaussian distribution on $[0, 1]$.

Per the previous section, we draw $n$ samples from each distribution. For a given quantile level $q \in (0, 1)$, we compute the empirical risk minimiser for both the strictly proper composite loss of Example 3, and its "partially capped", quantile-corrected version of Example 7. We use Gaussian-kernelised scorers, combined with the Nyström approximation to the kernel with $k = 500$ prototypes.

One we have an empirical risk minimiser $f_n^*$, we estimate its regret with respect to the partially capped loss with parameter $\alpha_q$, where $\alpha_q$ is the (explicitly computable) $q$th quantile of the density distribution $p(\mathsf{X})$. That is, we estimate $R(f_n^*) - R(f^*)$, where $f^*$ is the Bayes-optimal scorer $f^*(x) = p(x) \wedge \alpha_q$. This estimation is done via a separate empirical sample of instances drawn from $p$, and a uniform background.

Figures 6 and 7 show the regret of the empirical risk minimiser for various values of $q \in (0, 1)$. We observe that:

- as expected, as the number of samples increases, the regret decreases
- the regrets are generally higher for $q = 0.95$ than all other cases; this is consistent with our intuition that full density estimation is more challenging than partial density estimation

(a) $q = 0.25$.  (b) $q = 0.50$.  (c) $q = 0.75$.

Figure 5: Optimal solution for kernelised scorers, and various quantile levels $q \in (0, 1)$.

(a) `linear`

(b) `arcsin`

(c) `truncnorm`

Figure 6: Calibration regret of empirical risk minimiser for the strictly proper and partially proper composite loss, as a function of the number of training samples. Note that the vertical scales vary across the rows.

- the regrets are generally higher for $q = 0.05$ than $q = 0.50$; this possibly indicates that while the former involves an "easier" target object, one has to grapple with fewer available samples from the tail of the distribution

## D.3  Anomaly detection performance on real-world datasets

We finally evaluate the anomaly detection performance of various approaches on real-world datasets. We emphasise that standard (as opposed to calibrated) anomaly detection is *not* the primary goal of our partially proper loss approach; while we can use our predictions for this task, we expect there to be a slight dip in performance compared to bespoke approaches. Nonetheless, we illustrate here that our method is consistently competitive.

| (a) `linear` | (b) `arcsin` | (c) `truncnorm` |

Figure 7: Calibration regret of empirical risk minimiser for the partially proper composite loss, as a function of the number of training samples. On each plot, we show the regrets for different values of $q$, controlling the desired quantile. Note that the vertical scales vary across the rows.

| (a) Highest score. | (b) Lowest score. |

Figure 8: Samples of instances with highest and lowest score according to partially proper loss solution, `usps` dataset.

### D.3.1 Qualitative analysis: anomalous digits

To begin, we perform a qualitative evaluation of anomaly detection performance on the `usps` dataset. Following Schölkopf et al. [2001], we work with the provided test set, and augment each instance with a binary one-hot encoding of the corresponding digit. We fit our method on the resulting instances, and then examine those which are assigned lowest scores. Intuitively, these instances are expected to be visually anomalous, as they lie in a region of low density.

Figure 8 shows a sample of the instances with highest and lowest score. Each instance is annotated with the corresponding class label. The results are largely intuitive: most instances with low score are indeed visually anomalous.

### D.3.2 Quantitative analysis

We now provide a quantitative analysis of anomaly detection performance. For several real-world datasets, we treat specific instances as drawn from an anomalous distribution $P$, and others as drawn from a nominal (non-anomalous) distribution $\mu$. We provide as input to various methods a training sample of anomalous instances, and a desired quantile level $q \in (0, 1)$. The learned scorer $f \colon \mathcal{X} \to \mathbb{R}$ is applied to a testing sample comprising a mixture of anomalous and nominal instances. We evaluate the false negative and false positive rates of $f$ for the resulting binary classification task of distinguishing the nominal (positive) from anomalous (negative) instances, i.e., $P(f(\mathsf{X}) < 0)$ and $\mu(f(\mathsf{X}) > 0)$. Following Steinwart et al. [2005], we term these the *alarm* and *miss rates* respectively. Note that the provided quantile $q$ is meant to control the alarm rate.

We consider the following datasets:

- `MoU`. This synthetically generated data was constructed per Steinwart et al. [2005]. Here, $\mathcal{X} \subset \mathbb{R}^{10}$, with $\mathsf{X} = \mathbf{A}\mathsf{U}$ for $\mathsf{U} \sim \mathrm{Uniform}([0, 1]^{27})$, and $\mathbf{A} \in \mathbb{R}^{10 \times 27}$. For each row of $\mathbf{A}$,

Figure 9: Miss-alarm curves for various real-world datasets. Note that the scales vary across plots. Note also that the `usps` dataset has a different range of alarm rates.

we choose a uniformly random number $m \in \{2, 3, 4, 5\}$ of non-zero entries with value $\frac{1}{m}$. We draw nominal samples from $\mathrm{Uniform}([0,1]^{10})$.

- `MoG`. This synthetically generated data was constructed per Chen et al. [2013]. Here,

$$\mathsf{X} \sim 0.2 \cdot \mathcal{N}\left(\begin{bmatrix} 5 \\ 0 \end{bmatrix}, \begin{bmatrix} 1 & 0 \\ 0 & 9 \end{bmatrix}\right) + 0.8 \cdot \mathcal{N}\left(\begin{bmatrix} -5 \\ 0 \end{bmatrix}, \begin{bmatrix} 9 & 0 \\ 0 & 1 \end{bmatrix}\right).$$

The reference measure $\mu$ is the uniform distribution over $[-18, +18]^2$, from which we draw nominal samples.

- `satellite`. This dataset is taken from the UCI repository. Following Chen et al. [2013], we treat the three smallest classes as anomalous, and all other classes as nominal.
- `usps`. This dataset is taken from the UCI repository. Following Schölkopf et al. [2001], we treat the digit zero as anomalous, and all other digits as nominal.

Figure 9 summarises how the test set miss rate changes as we request different alarm rates, corresponding to the quantile levels $q$. We see that in general, our partial density estimation approach can achieve competitive or lower miss rates than full density estimation or the OC-SVM. We emphasise that, per §6.2, our framework when instantiated with the LSIF loss is in fact closely related to the OC-SVM; thus, the similar performance of the two methods is to be expected.

# E  Calibrated anomaly detection and entropy

The problem of calibrated anomaly detection has a surprising relationship to the estimation of a particular entropy of the distribution $P$. We make use of the following result [Reid and Williamson, 2011, Theorems 9, 18]: Suppose $f\colon \mathbb{R}_{\geq 0} \to \mathbb{R}$ is a convex function normalised such that $f(1) = 0$, and $P$ and $Q$ be two probability distributions on $\mathfrak{X}$. Let $I_f(P,Q) = \int_{\mathfrak{X}} f\left(\frac{\mathrm{d}P}{\mathrm{d}Q}\right) \mathrm{d}Q$ denote the Csizár $f$-divergence between $P$ and $Q$. Let $\Delta \underline{\mathbb{L}}_\ell(\pi, P, Q)$ denote the *statistical information* (with respect to $\ell$) of the binary experiment with class conditional distributions $P$ and $Q$ and prior probability $\pi$ (i.e. the difference between the prior and posterior risk; see Reid and Williamson [2011]). Then for any given $\ell$, there exists an $f$ such that for all $P, Q$

$$I_f(P,Q) = \Delta\underline{\mathbb{L}}_\ell(1/2, P, Q). \tag{29}$$

The particular $f$, given $\ell$ can be expressed conveniently in terms of the corresponding weight functions: $w$ for $\ell$ and $\gamma$ for $f$, where

$$\gamma(c) = \frac{1}{c^3} \cdot f''\left(\frac{1-c}{c}\right), \quad c \in (0,1), \tag{30}$$

or equivalently

$$f(s) = \int_0^s \underbrace{\left(\int_0^t \frac{1}{(\tau+1)^3} \gamma\left(\frac{1}{\tau+1}\right) \mathrm{d}\tau\right)}_{\phi(t)} \mathrm{d}t. \tag{31}$$

The correspondence we need is simply

$$\gamma(c) = \frac{1}{16} \cdot w(1-c), \quad c \in [0,1]. \tag{32}$$

When considering the anomaly detection problem, we replace $Q$ by $\mu$ and consider it fixed. Thus consider the function

$$H_f^\mu(P) := I_f(P, \mu), \tag{33}$$

which is known as the Csiszár $f$-entropy of $P$ with respect to reference measure $\mu$.

As explicated in Reid and Williamson [2011], the $f$-divergence $I_f(P, \mu)$ can be computed in a variational form:

$$I_f(P,\mu) = \sup_{\rho\colon \mathfrak{X} \to \mathbb{R}} \mathop{\mathbb{E}}_P [\rho] - \mathop{\mathbb{E}}_\mu [f^* \circ \rho], \tag{34}$$

where $f^*$ is the Legendre-Fenchel conjugate of $f$, and the supremum is over all measurable functions from $\mathfrak{X}$ to $\mathbb{R}$.

In the main body of the paper we constructed a $c_\alpha$-strictly proper loss $\bar{\ell}$ from a strictly proper loss $\ell$ by transforming the weight functions according to

$$\bar{w}(c) = [\![c \leq c_\alpha]\!] \cdot w(c), \quad c \in [0,1]. \tag{35}$$

If $\ell$ corresponds to $f$ via its weight function $\gamma$ according to (32), then the weight function for $\bar{f}$ corresponding to $\bar{\ell}$ is given by

$$\bar{\gamma}(c) = [\![c > 1 - c_\alpha]\!] \cdot \gamma(c). \tag{36}$$

Combining (36) with (31) we see that the corresponding $\bar{\phi}$ satisfies

$$\bar{\phi}(s) = \phi(s) \wedge \phi(s^*) \tag{37}$$

and thus $\bar{f}(s) = \bar{f}_1(s) + \bar{f}_2(s)$ for $s \in \mathbb{R}_{\geq 0}$, where

$$\bar{f}_1(s) = [\![s \leq s^*]\!] \cdot f(s) \tag{38}$$

$$\bar{f}_2(s) = [\![s > s^*]\!] \cdot (f'(s^*) \cdot s + f(s^*) - f'(s^*)s^*), \tag{39}$$

where $s^* = \frac{1}{1-c_\alpha} - 1$. That is $\bar{f}$ equals $f$ on the interval $[0, s^*]$ and then affinely continues. Consequently

$$H_{\bar{f}}^\mu(P) = \int_{\mathfrak{X}} f\left(\frac{\mathrm{d}P}{\mathrm{d}\mu}\right) \mathrm{d}\mu$$

$$= \int\limits_{\{x\in\mathcal{X}\mid\frac{\mathrm{d}P}{\mathrm{d}\mu}(x)<f'(s^*)\}} f\left(\frac{\mathrm{d}P}{\mathrm{d}\mu}\right)\mathrm{d}\mu + \int\limits_{\{x\in\mathcal{X}\mid\frac{\mathrm{d}P}{\mathrm{d}\mu}(x)\geq f'(s^*)\}} \left(f'(s^*)\frac{\mathrm{d}P}{\mathrm{d}\mu}(x) + f(s^*) - f'(s^*)s^*\right)\mathrm{d}\mu(x)$$

$$= \int_{\mathcal{X}} f\left(\widetilde{\frac{\mathrm{d}P}{\mathrm{d}\mu}}\right)\mathrm{d}\mu + \underbrace{f'(s^*) + f(s^*) - f'(s^*)s^*}_{\text{constant}},$$

where

$$\widetilde{\frac{\mathrm{d}P}{\mathrm{d}\mu}}(x) = \begin{cases} \frac{\mathrm{d}P}{\mathrm{d}\mu}(x), & \frac{\mathrm{d}P}{\mathrm{d}\mu}(x) < f'(s^*) \\ 0 & \text{otherwise} \end{cases}$$

is the density of the low density region only.

By properties of the Legendre-Fenchel conjugate, we have

$$\bar{f}^*(v) = \begin{cases} f^*(v) & v < f'(s^*) \\ +\infty, & v^* \geq f'(s^*) \end{cases}$$

and thus the variational representation (34) can be rewritten as

$$H_{\bar{f}}^{\mu}(P) = \sup_{\rho:\, \mathcal{X}\to(-\infty, f'(s^*)]} \mathbb{E}_P[\rho] - \mathbb{E}_{\mu}[f^* \circ \rho]. \tag{40}$$

That is, the reweighting from $f$ to $\bar{f}$ is implemented by merely restricting the range of functions over which one optimises. The argmax in (40) corresponds via simple transformation to the Bayes optimal hypothesis in the risk minimization problem (Proposition 4).

We have thus seen that the problem of calibrated anomaly detection of $P$ relative to $\mu$ using $\bar{\ell}$ is equivalent to the determination of the $\bar{f}$-entropy $H_{\bar{f}}^{\mu}(P)$ of $P$ relative to $\mu$.