[Reviews · NeurIPS 2018]

Reviewer 1



Summary of the article The article “A loss framework for calibrated anomaly detection” extends the traditional idea of anomaly detection—detecting points in low-density regions—to include a measure of confidence of the anomalousness of a point. The work is built atop a classification framework for the density sublevel set problem, theoretically demonstrating that target objective of calibrated anomaly detection can be accomplished by minimizing risk in a class of loss function. The loss functions have a relationship with pinball loss, the minimization can be done relatively efficiently for certain functions, and connections with one-class SVM are also established. In addition to the calibration and (sometimes) tractable optimization, the framework allows for classifications that are bayes-optimal and have quantile control. Review This article is very well written and was extremely enjoyable to read. I find that the authors clearly and succinctly layout their challenges and demonstrate their proposed solutions. I have no major criticisms of the work. I would, however, argue that the authors could increase the practical implications of their work by providing a simulation study comparing their framework to at least the other loss-based approaches to anomaly detection. The one-class svm is a popular and commonly used anomaly detection technique, that has been shown to perform well. Therefore, a finite sample comparison (to contrast the asymptotic theory) would provide practitioners a valuable understanding. After Author Response The authors provided some experimental results as I requested, and it does complement their theory. I think experiments are also helpful for highlighting when the algorithm breaks/down has poor performance; this is also typically a valuable complement to theory. This is, however, minor and I think this paper is deservant of acceptance.

Reviewer 2



The paper is of very high quality and clarity. I includes a well-written background and summarization of approaches, placing this work in calibrated anomaly detection. The authors apparently speak math more fluently than this reviewer, nevertheless, because of careful highlighting, and with a little commitment, this is accessible to readers outside of the primary subject area. This paper provides an interesting theoretical contribution. However, there is no experiment or application, as is often the case for strong NIPS papers that relate theory to practice. Minor comments: Figure 1: The first term (density estimation in Section 3.2) appears to be missing square brackets in the expectation

Reviewer 3



This paper studies the problem of calibrated anomaly detection by introducing a new type of loss functions that may depend on the density threshold. By incorporating quantile control, the authors develop algorithms that can estimate the target function and the score threshold simultaneously. The author also discuss how to solve the corresponding optimization problem based on partially capped loss and a kernel absorption trick. The proposed approach includes one-class SVM as a special case. The paper is well written and easy to read. I am just wondering if it’s possible to give some consistency results or generalization error bounds? The experimental results looks promising.